# Pathologies and Challenges of Using Differentiable Simulators in Policy Optimization for Contact-Rich Manipulation

H.J. Terry Suh, Max Simchowitz, Kaiqing Zhang, Tao Pang, Russ Tedrake

*Abstract*—**Policy search methods in Reinforcement Learning (RL) have shown impressive results in contact-rich tasks such as dexterous manipulation. However, the high variance of zero-order Monte-Carlo gradient estimates results in slow convergence and a requirement for a high number of samples. By replacing these zero-order gradient estimates with first-order ones, differentiable simulators promise faster computation time for policy gradient methods when the model is known. Contrary to this belief, we highlight some of the pathologies of using first-order gradients and show that in many physical scenarios involving rich contact, using zero-order gradients result in better performance. Building on these pathologies and lessons, we propose guidelines for designing differentiable simulators, as well as policy optimization algorithms that use these simulators. By doing so, we hope to reap the benefits of first-order gradients while avoiding the potential pitfalls.**

## I. INTRODUCTION

Reinforcement Learning (RL) is fundamentally concerned with the problem of minimizing a *stochastic objective*,

$$\min_{\boldsymbol{\theta}} F(\boldsymbol{\theta}) = \min_{\boldsymbol{\theta}} \mathbb{E}_{\mathbf{w}} f(\boldsymbol{\theta}, \mathbf{w}).$$

Many algorithms in RL heavily rely on *zeroth-order* Monte-Carlo estimation of the gradient $\nabla F$ [27, 22]. Yet, in contact-rich robotic manipulation where we have model knowledge and structure of the dynamics, it is possible to differentiate through the physics and obtain *exact* gradients of $f$, which can also be used to construct a *first-order* estimate of $\nabla F$. The availability of both options begs the question: given access to gradients of $f$, which estimator should we prefer?

In stochastic optimization, the theoretical benefits of using first-order estimates of $\nabla F$ over zeroth-order ones have mainly been understood through the lens of variance and convergence rates [10, 16]: the first-order estimator often (*not always*) results in much less variance compared to the zeroth-order one, which leads to faster convergence rates to a local minima of nonconvex smooth objective functions. However, the landscape of RL objectives that involve long-horizon sequential decision making (e.g. policy optimization) is challenging to analyze, and convergence properties in these landscapes are relatively poorly understood. In particular, contact-rich systems can display complex characteristics including nonlinearities, non-smoothness, and discontinuities (Figure 1) [29, 17, 25].

Nevertheless, lessons from convergence rate analysis tell us that there may be benefits to using the exact gradients even for these complex physical systems. Such ideas have been championed through the term "differentiable simulation", where forward simulation of physics is programmed in a manner that is consistent with automatic differentiation [8, 12, 28, 30, 9], or computation of analytic derivatives [3]. These methods

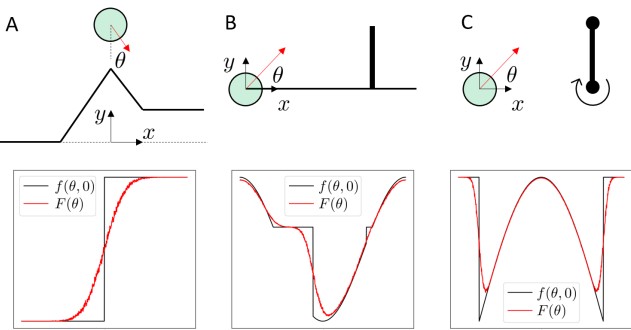

Fig. 1. Examples of simple optimization problems on physical systems. Goal is to: A. maximize $y$ position of the ball after dropping. B. maximize distance thrown, with a wall that results in inelastic impact. C. maximize transferred angular momentum to the pivoting bar through collision. Second row: the original objective and the stochastic objective after randomized smoothing.

have shown promising results in decreasing computation time compared to zeroth-order methods [13, 8, 11, 6, 5, 19].

However, due to the complex characteristics of contact dynamics, we show that the belief that first-order gradients improve performance over zero-order ones is not always true for contact-rich manipulation. We illustrate this phenomenon through couple pathologies: first, even under sufficient regularity conditions of continuity, the choice of contact modeling can cause the first-order gradient estimate to have higher variance compared to the zeroth-order one. In particular, this may occur in approaches that utilize the penalty method [14], which requires stiff dynamics to realistically simulate contact [9].

In addition, we show that many contact-rich systems display nearly/strictly *discontinuous* behavior in the underlying landscape. The presence of such discontinuities causes the first-order gradient estimator to be *biased*, while the zeroth-order one still remains unbiased. Furthermore, we show that even when continuous approximations are made, such approximations are often stiff and highly-Lipschitz. In these settings, the first order estimator still suffer from what we call *empirical bias* under finite-sample settings. The compromise of the first order estimator in the face of more accurate description of contact dynamics hints at a fundamental tension between realism of the dynamics and the performance of first-order gradients.

From these pathologies, we suggest methods in simulation, as well as algorithms, that may improve the efficacy of first-order gradient estimates obtained using differentiable simulation. We advocate for the use of implicit contact models that are less stiff, and thus have low variance of the first-order gradient. In addition, we show they can be analytically smoothed out to mitigate discontinuities. Finally, we introduce a method to interpolate gradients that escapes these identified pitfalls.

## II. PRELIMINARIES

### A. Policy Optimization Setting

We study a discrete-time, finite-horizon, continuous-state control problem with states $\mathbf{x} \in \mathbb{R}^n$, inputs $\mathbf{u} \in \mathbb{R}^m$, transition function $\phi : \mathbb{R}^n \times \mathbb{R}^m \to \mathbb{R}^n$, and horizon $H \in \mathbb{N}$. Given a sequence of costs $c_h : \mathbb{R}^n \times \mathbb{R}^m \to \mathbb{R}$, a family of policies $\pi_h(\cdot, \cdot) : \mathbb{R}^n \times \mathbb{R}^d \to \mathbb{R}^m$ parameterized by $\boldsymbol{\theta} \in \mathbb{R}^d$, and a sequence of injected noise terms $\mathbf{w}_{1:H} \in (\mathbb{R}^m)^H$, we define the cost-to-go functions

$$V_h(\mathbf{x}_h, \mathbf{w}_{h:H}, \boldsymbol{\theta}) = \sum_{h'=h}^{H} c_h(\mathbf{x}_{h'}, \mathbf{u}_{h'}),$$

$$\text{s.t. } \mathbf{x}_{h'+1} = \phi(\mathbf{x}_{h'}, \mathbf{u}_{h'}), \ \mathbf{u}_{h'} = \pi(\mathbf{x}_{h'}, \boldsymbol{\theta}) + \mathbf{w}_{h'}, \ h' \geq h.$$

Our aim is to minimize the policy optimization objective

$$F(\boldsymbol{\theta}) := \mathbb{E}_{\mathbf{x}_1 \sim \rho} \mathbb{E}_{\mathbf{w}_h \overset{\text{i.i.d.}}{\sim} p} V_1(\mathbf{x}_1, \mathbf{w}_{1:H}, \boldsymbol{\theta}), \quad (1)$$

where $\rho$ is a distribution over initial states $\mathbf{x}_1$, and $\mathbf{w}_1, \ldots, \mathbf{w}_H$ are i.i.d. according to $p$ which we assume to be a zero-mean Gaussian with covariance $\sigma^2 I_n$.

### B. Zeroth-order estimator.

The policy gradient can be estimated only using samples of the function values [31].

**Definition II.1.** Given a single zeroth-order estimate of the policy gradient $\hat{\nabla}^{[0]} F_i(\boldsymbol{\theta})$, we define the zeroth-order batched gradient (ZoBG) $\bar{\nabla}^{[0]} F(\boldsymbol{\theta})$ as the sample mean,

$$\hat{\nabla}^{[0]} F_i(\boldsymbol{\theta}) := \frac{1}{\sigma^2} V_1(\mathbf{x}_1, \mathbf{w}_{1:H}^i, \boldsymbol{\theta}) \left[ \sum_{h=1}^{H} \mathrm{D}_{\boldsymbol{\theta}} \pi(\mathbf{x}_h^i, \boldsymbol{\theta})^{\mathsf{T}} \mathbf{w}_h^i \right]$$

$$\bar{\nabla}^{[0]} F(\boldsymbol{\theta}) := \frac{1}{N} \sum_{i=1}^{N} \hat{\nabla}^{[0]} F_i(\boldsymbol{\theta}),$$

where $\mathbf{x}_h^i$ is the state at time $h$ of a trajectory induced by the noise $\mathbf{w}_{1:H}^i$, $i$ is the index of the sample trajectory, and $\mathrm{D}_{\boldsymbol{\theta}} \pi$ is the Jacobian matrix $\partial \pi / \partial \boldsymbol{\theta} \in \mathbb{R}^{m \times d}$.

The hat notation denotes a per-sample Monte-Carlo estimate, and bar-notation a sample mean. The ZoBG is also referred to as the REINFORCE [31], score function, or the likelihood-ratio gradient. In practice, a baseline term $b$ is subtracted from $V_1(\mathbf{x}_1, \mathbf{w}_{1:H}^i, \boldsymbol{\theta})$ for variance reduction. One example is the zero-noise rollout as the baseline $b = V_1(\mathbf{x}_1, \mathbf{0}_{1:H}, \boldsymbol{\theta})$:

### C. First-Order Estimator.

In differentiable simulators, the gradients of the dynamics $\phi$ and costs $c_h$ are available *almost surely* (i.e., with probability one). Hence, one may compute the exact gradient $\nabla_{\boldsymbol{\theta}} V_1(\mathbf{x}_1, \mathbf{w}_{1:H}, \boldsymbol{\theta})$ by automatic differentiation and average them to estimate $\nabla F(\boldsymbol{\theta})$.

**Definition II.2.** Given a single first-order gradient estimate $\hat{\nabla}^{[1]} F_i(\boldsymbol{\theta})$, we define the first-order batched gradient (FoBG) as the sample mean:

$$\hat{\nabla}^{[1]} F_i(\boldsymbol{\theta}) := \nabla_{\boldsymbol{\theta}} V_1(\mathbf{x}_1, \mathbf{w}_{1:H}^i, \boldsymbol{\theta})$$

$$\bar{\nabla}^{[1]} F(\boldsymbol{\theta}) := \frac{1}{N} \sum_{i=1}^{N} \hat{\nabla}^{[1]} F_i(\boldsymbol{\theta}).$$

The FoBG is also referred to as the reparametrization gradient [15], the pathwise derivative [21], or Back Propagation through Time (BPTT).

## III. PITFALLS OF FIRST-ORDER GRADIENTS

In this section, we shows pathologies in contact-rich systems for which the FoBG can perform worse than the ZoBG.

### A. Bias under discontinuities

Under standard regularity conditions, it is well-known that both estimators are unbiased estimators of the true gradient $\nabla F(\boldsymbol{\theta})$. However, care must be taken to define these conditions precisely, as such conditions are broken for contact-rich systems. Fortunately, the ZoBG is still unbiased under mild assumptions,

$$\mathbb{E}[\bar{\nabla}^{[0]} F(\boldsymbol{\theta})] = \nabla F(\boldsymbol{\theta}).$$

In contrast, the FoBG requires strong continuity conditions in order to satisfy the requirement for unbiasedness. However, under Lipschitz continuity, it is indeed unbiased.

**Lemma III.1.** *If $\phi(\cdot, \cdot)$ is locally Lipschitz and $c_h(\cdot, \cdot) \in C^\infty$, then $\bar{\nabla}^{[1]} F(\boldsymbol{\theta})$ is defined almost surely, and*

$$\mathbb{E}[\bar{\nabla}^{[1]} F(\boldsymbol{\theta})] = \nabla F(\boldsymbol{\theta}).$$

Lemma III.1 tells us that FoBG can fail when applied to discontinuous landscapes. We illustrate a simple case of biasedness through a counterexample.

*Example* III.2 (**Heaviside**). [2, 25] Consider the Heaviside function,

$$f(\boldsymbol{\theta}, \mathbf{w}) = H(\boldsymbol{\theta} + \mathbf{w}), \quad H(t) = \mathbb{1}_{t \geq 0}$$

whose stochastic objective becomes the error function

$$F(\boldsymbol{\theta}) = \mathbb{E}_{\mathbf{w}}[H(\boldsymbol{\theta} + \mathbf{w})] = \text{erf}(-\boldsymbol{\theta}; \sigma^2),$$

However, since $\nabla_{\boldsymbol{\theta}} H(\boldsymbol{\theta} + \mathbf{w}) = 0$ for all $\boldsymbol{\theta} \neq -\mathbf{w}$, we have $\mathbb{E}_{\mathbf{w}_i} \delta(\boldsymbol{\theta} + \mathbf{w}_i) = 0$. Hence, the Law of Large Numbers does not hold, and FoBG is biased as the gradient of the stochastic objective, a Gaussian, is non-zero at any $\boldsymbol{\theta}$. We further note that the empirical variance of the FoBG estimator in this example is zero. On the other hand, the ZoBG escapes this problem and provides an unbiased estimate, since it always takes finite intervals that include the integral of the delta.

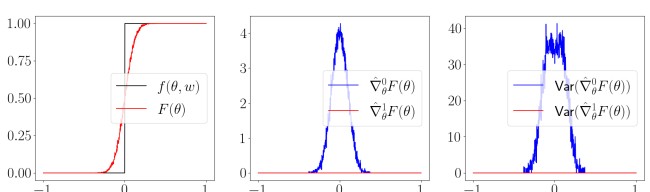

Fig. 2. From left: heaviside objective $f(\boldsymbol{\theta}, \mathbf{w})$ and stochastic objective $F(\boldsymbol{\theta})$, empirical values of the gradient estimates, and their empirical variance.

### B. The "Empirical bias" phenomenon

One might argue that *strict* discontinuity is simply an artifact of modeling choice in simulators; indeed, many simulators approximate discontinuous dynamics as a limit of continuous ones with growing Lipschitz constant [9, 7]. In this section, we explain how this can lead to a phenomenon we call *empirical bias*, where the FoBG appears to have low empirical variance,

but is still highly inaccurate; i.e. it "looks" biased when a finite number of samples are used. Through this phenomenon, we claim that performance degradation of first-order gradient estimates do not require strict discontinuity, but is also present in continuous, yet *stiff* approximations of discontinuities.

**Definition III.3** (Empirical bias). Let $\mathbf{z}$ be a vector-valued random variable with $\mathbb{E}[\|\mathbf{z}\|] < \infty$. We say $\mathbf{z}$ has $(\beta, \Delta, S)$-empirical bias if there is a random event $\mathcal{E}$ such that $\Pr[\mathcal{E}] \geq 1 - \beta$, and $\|\mathbb{E}[\mathbf{z} \mid \mathcal{E}] - \mathbb{E}[\mathbf{z}]\| \geq \Delta$, but $\|\mathbf{z} - \mathbb{E}[\mathbf{z} \mid \mathcal{E}]\| \leq S$ almost surely on $\mathcal{E}$.

A paradigmatic example of empirical bias is a random scalar $\mathbf{z}$ which takes the value $0$ with probability $1 - \beta$, and $\frac{1}{\beta}$ with probability $\beta$. Setting $\mathcal{E} = \{\mathbf{z} = 0\}$, we see $\mathbb{E}[\mathbf{z}] = 1$, $\mathbb{E}[\mathbf{z} \mid \mathcal{E}] = 0$, and so $\mathbf{z}$ satisfies $(\beta, 1, 0)$-empirical bias. Note that $\mathbf{Var}[\mathbf{z}] = 1/\beta - 1$; in fact, small-$\beta$ empirical bias implies large variance more generally.

**Lemma III.4.** *Suppose $\mathbf{z}$ has $(\beta, \Delta, S)$-empirical bias. Then* $\mathbf{Var}[\mathbf{z}] \geq \frac{\Delta_0^2}{\beta}$, *where* $\Delta_0 := \max\{0, (1 - \beta)\Delta - \beta\|\mathbb{E}[\mathbf{z}]\|\}$.

Empirical bias naturally arises for discontinuities or stiff continuous approximations.

*Example* III.5 (**Coulomb friction**). The Coulomb model of friction is discontinuous in the relative tangential velocity between two bodies. In many simulators [9, 4], it is common to consider a continuous approximation instead. We idealize such approximations through a piecewise linear relaxation of the Heaviside that is continuous, parametrized by the width of the middle linear region $\nu$ (which corresponds to *slip tolerance*).

$$\bar{H}_\nu(t) = \begin{cases} 2t/\nu & \text{if } |t| \leq \nu/2 \\ 2H(t) - 1 & \text{else} \end{cases}.$$

In practice, lower values of $\nu$ lead to more realistic behavior in simulation [28], but this has adverse effects for empirical bias. Considering $f_\nu(\boldsymbol{\theta}, \mathbf{w}) = \bar{H}_\nu(\boldsymbol{\theta} + \mathbf{w})$, we have $F_\nu(\boldsymbol{\theta}) = \mathbb{E}_\mathbf{w}[\bar{H}_\nu(\boldsymbol{\theta} + \mathbf{w})] := \text{erf}(\nu/2 - \theta; \sigma^2)$. In particular, setting $c_\sigma := \frac{1}{\sqrt{2\pi}\sigma}$, then at $\boldsymbol{\theta} = \nu/2$, $\nabla F_\nu(\boldsymbol{\theta}) = c_\sigma$, whereas, with probability at least $c_\sigma \nu$, $\nabla f_\nu(\boldsymbol{\theta}, \mathbf{w}) = 0$. Hence, the FoBG has $(c_\sigma \nu, c_\sigma, 0)$ empirical bias, and its variance scales with $1/\nu$ as $\nu \to 0$. The limiting $\nu = 0$ case, corresponding to the Coulomb model, is the Heaviside from Example III.2, where the limit of high empirical bias, as well as variance, becomes biased in expectation (but, surprisingly, zero variance!). We empirically illustrate this effect in Figure 3. We also note that more complicated models of friction (e.g. that incorporates the Stribeck effect [24]) would suffer similar problems.

*Example* III.6 (**Discontinuity in geometry**). Another source of discontinuity in simulators comes from the discontinuity of surface normals. We show this in Figure 4, where balls that collide with a rectangular geometry create discontinuities. It is possible to make a continuous relaxation [7] by considering a smoother geometry, depicted by the addition of the dome in Figure 4. While this makes FoBG no longer biased asymptotically, the stiffness of the relaxation still results in high empirical bias.

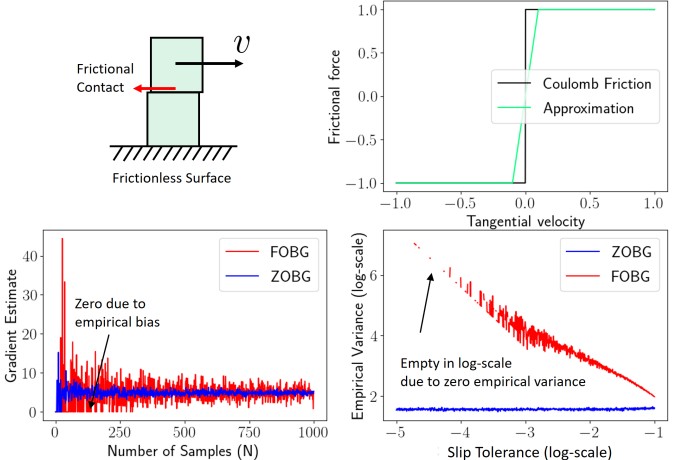

Fig. 3. Top column: illustration of the physical system and the relaxation of Coulomb friction. Bottom column: the values of estimators and their empirical variances depending on number of samples and slip tolerance. Values of FoBG are zero in low-sample regimes due to empirical bias. As $\nu \to 0$, the empirical variance of FoBG goes to zero, which shows as empty in the log-scale. Expected variance, however, blows up as it scales with $1/\nu$.

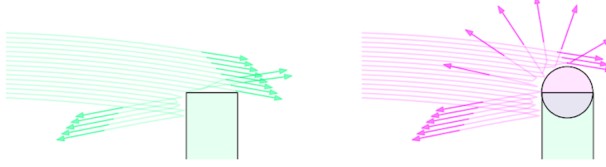

Fig. 4. Left: example of ball hitting the wall. The green trajectories hit a rectangular wall, displaying discontinuities. Right: the pink trajectories collide with the dome on top, and show continuous but stiff behavior.

### C. High Variance from Stiffness

Even without the phenomenon of empirical bias, we show that certain choices of contact models can cause the FoBG to suffer from high variance. In particular, approximations of rigid contact with high-stiffness spring models (i.e. penalty method) causes the gradient may have a high norm.

*Example* III.7. (**Pushing with stiff contact**). We demonstrate this phenomenon through a simple 1D pushing example in Figure 5, where the ZoBG has lower variance than the FoBG as stiffness increases, until numerical semi-implicit integration becomes unstable under a fixed timestep.

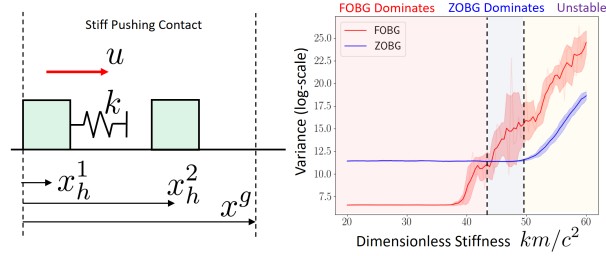

Fig. 5. The variance of the gradient of $V_1$, with running cost $c_h = \|\mathbf{x}_h^2 - \mathbf{x}^g\|^2$, with respect to input trajectory as spring constant $k$ increases. Mass $m$ and damping coefficient $c$ are fixed.

## IV. Tackling the Pathologies: A Path Forward

In this section, we comment on methods that can alleviate the pathologies that were found in the previous section.

### A. Less Stiff Formulations of Contact Dynamics

In order to avoid high variance of the FoBG, we must ensure that the norm of the gradient is low. Yet, as illustrated by Example III.7., approximating contact using stiff springs, as done in works that model contact with the *penalty method*, inevitably results in trading off stiffness and physical realism.

Therefore, we advocate less stiff contact models that are based on implicit time-stepping [23], whose per time-step computation relies on solving optimization problems such as the Linear Complementary Problem (LCP), which can be further relaxed into solving convex Quadratic Programs (QP)s [1]. The derivatives of such systems can be obtained by the implicit function theorem, differentiating through the optimality conditions of the problems. We give one example of such a convex QP as below. Correctly using gradients from implicit time-stepping can vastly improve the efficacy of FoBG by ensuring that their norm stays reasonably bounded.

*Example* IV.1. (**Implicit Time-Stepping for Pushing**). We illustrate implicit time-stepping with a 1-dimensional example consisting of a point mass and a wall. The state of the system is $(x, v) \in \mathbb{R}^2$, where $x$ is the position and $v$ the velocity of the point mass. The non-penetrable wall occupies $x \leq 0$.

The equations of motion of the system is

$$m(v_+ - v) = u + \lambda, \quad (2a)$$
$$x_+ = x + hv_+, \quad (2b)$$
$$0 \leq x_+ \perp \lambda \geq 0, \quad (2c)$$

where $(x_+, v_+)$ represent the system state at the next time step; $h$ is the step size; $m$ is the mass; $u$ is the impulse applied to the point mass by actuation; and $\lambda$ is the impulse due to contact with wall. Constraint (2a) is the momentum balance of the point mass. Constraint (2c) is the complementarity constraint that ensures the wall can only push on the point mass when they are in contact. We can indeed see that the equations of motion (2) is the KKT condition of the following QP:

$$\underset{v_+}{\text{minimize}} \quad \tfrac{1}{2}m(v_+ - v)^2 - uv_+ \quad (3a)$$
$$\text{subject to} \quad \frac{x}{h} + v_+ \geq 0 \quad (3b)$$

### B. Smooth Analytic Approximations of Dynamics

Although we show that strict discontinuity is not required to have degradation of performance for the FoBG, soft relaxations of discontinuities still behave much better. To this end, we also advocate for analytically providing a smooth surrogates of the discontinuous dynamics in simulation, and increasingly lowering the relaxation during the policy optimization step. To overcome the pathologies of using FoBGs, we believe that providing such a feature should be a *requirement* for differentiable simulators for them to be useful in policy optimization.

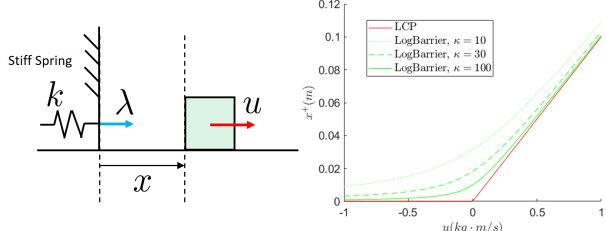

Fig. 6. Left: Visualization of wall and block examples in Example IV.1 and Example IV.2. Note that both schemes *do not* require using the spring constant $k$, where as the penalty method will. This alleviates problems associated with stiffness of the gradients. Right: Results of simulating the methods of Example IV.1 and Example IV.2 at $(x, v) = 0$. The resulting positions $x^+$ are plotted as functions of input impulse $u$.

Previous works have provided smooth surrogates to the penalty method of contact [9, 13, 32], which reasonably addresses discontinuities, yet still suffers from stiffness. Instead, we show that a smooth approximation can be made to implicit time-stepping methods by using common constraint relaxation methods such as the log-barrier function used in interior-point method.

*Example* IV.2. (**Smooth Relaxation for Pushing**). The optimization-based dynamics of Example IV.1 can be smoothed by replacing the non-penetration constraint (3b) with an additional log-barrier term in the objective (3a):

$$\underset{v_+}{\text{minimize}} \quad \tfrac{1}{2}m(v_+ - v)^2 - uv_+ - \tfrac{1}{\kappa}\log(\tfrac{x}{h} + v_+), \quad (4)$$

which is an unconstrained convex optimization program, whose optimality condition can be obtained by setting the derivative of the objective (4) to 0:

$$m(v_+ - v) = u + [\kappa (x/h + v_+)]^{-1}. \quad (5)$$

The optimality condition (5) can be interpreted as the momentum balance of the point mass, but the wall now acts as a force field, exerting on the object a force whose magnitude is inversely proportional to the distance to the wall. The strength of the force field is controlled by the log-barrier weight $\kappa$. As $\kappa \to \infty$, the solution of (4) converges to that of (3).

### C. Gradient Interpolation

Finally, we mention some recent advances on the algorithm side. If we can compute both the FoBG and the ZoBG using uncorrelated samples, we can consider an interpolated gradient,

$$\hat{\nabla}^{[\alpha]} F_i(\boldsymbol{\theta}) \coloneqq \alpha\hat{\nabla}^{[0]} F_i(\boldsymbol{\theta}) + (1-\alpha)\hat{\nabla}^{[1]}, F_i(\boldsymbol{\theta}) \quad (6)$$

where $\alpha \in [0, 1]$. Previous works on gradient interpolation [20, 18] shows that we can optimally interpolate the two gradients based on computing empirical variance. However, as Example III.2 shows, the empirical variance can be an unreliable estimate if FoBG is biased under discontinuities.

To mitigate this problem, we can test the correctness of the FoBG against the unbiased ZoBG by constructing a confidence interval based on samples of the ZoBG, and choosing an optimal value of $\alpha$ subject to a chance constraint on the allowable value of the interpolated gradient [26].

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
