# OpenReview forum: "Pathologies and Challenges of Using Differentiable Simulators in Policy Optimization for Contact-Rich Manipulation"
_ICRA.org/2022/Workshop/Contact-Rich — ICRA 2022 Workshop: RL for Manipulation Oral_

### Official Review · Reviewer_C26M · 2022-05-05

**Rating:** 8
**Confidence:** 3

**Review:**

### Summary
This paper presents guidelines for designing differentiable simulators and RL algorithms that rely on Monte-Carlo gradient estimates. This work discusses the advantages and limitations of using a first-order gradient estimate over a zeroth-order.

The topic of this paper is interesting and relevant to the workshop.

### Comments

- This paper is well organized and properly references the related work.
- This paper provides a well-elaborated discussion on the topic of gradient-based policy optimization methods, described in the summary.

---

### Official Review · Reviewer_Dh9W · 2022-05-08
**A nice study on using zero- vs. first-order gradients in policy learning that can be improved by adding qualitative evaluations of the resulting policies**

**Rating:** 8
**Confidence:** 5

**Review:**

This paper studies the use of differentiable simulators for gradient computations in policy optimization for contact-rich manipulation tasks.
Such simulators can provide first-order gradients in contrast to classical zero-order gradient estimation using samples.
The authors conduct various case studies to highlight the advantages and disadvantages of each gradient approach in various example tasks.
For instance, first-order gradients often result in less variance than zero-order ones but might result in biased estimators.
The paper finishes by summarizing ideas to combine the benefits of both gradient approaches in real systems.
This paper presents a very nice study on zero- and first-order gradients in policy optimization.
The authors nicely introduce and motivate problem and highlight why this topic requires more attention from the community.
The paper is written in a tutorial fashion, helping to guide readers through the study.
The authors introduce various example contact-rich manipulation tasks to investigate the gradient computations in more detail.
The examples are well chosen and intuitive to understand the differences for zero-order and first-order gradients.
Nevertheless, the paper can be further improved to communicate the challenges to an even broader audience.
This can be mainly done by simplifying explanations in the main section of the paper.
Although the paper motivates the problem in an easy-to-follow fashion, the analysis section is more difficult to follow for readers.
The paper often does not motivate certain modeling or computation steps enough.
For instance, why are the chosen assumptions for ZoBG mild (Sec. IIIA)? Why does the Law of Large Numbers not hold anymore (Sec. IIIA)? Why are the motion equations in Eq. 2 the KKT condition?
The Coulomb friction example is very important, but the explanations are difficult to follow to the various steps. Can the authors simplify this part and illustrate it?
A large part of the study deals with biased estimators.
Although the core problem is motivated, the paper would have an added benefit if the authors can show what happens when one uses biased estimators.
The study so far is very quantitative but would benefit from using such gradients to provide qualitative insides on policy learning.
For instance, how biased can estimators be? How much better do policies get?
To save space, the paper can reduce certain definitions.
For instance, the introduced exact definitions of the gradients and the system overview can be shortened and referenced instead.
This reduction would provide more space to provide more figures to explain the content.
Lastly, Section IV (tackling the problems) could be extended by adding some proof-of-concept examples of addressing the challenges.
Such examples would nicely close the loop of the tutorial and provide readers a well-rounded explanation, analysis, and possible solutions of the problems.
Considering the style, the paper should use the regular IEEE bib style to sort numbering in an ascending order. This ordering makes it easier to cross-reference papers.
The bib file also needs a revision, e.g., [9] and [11] miss conference information.
Also, the use of balancing the columns destroyed the formatting: [15] and [23] are mixed up.
Some figures miss proper axes labels, e.g., Fig. 2.
To improve readability of the figures for visually impaired, the authors are encouraged to avoid the sole use of red and green colors.
The paper's readability can also be improved by using equation numbering more consistently.
In summary, this paper presents a nice study that is highly relevant to the workshop's core idea.
The paper's tutorial style can be further improved by simplifying the main sections and providing more qualitative comparisons.